# Perceptions of Multistakeholder Partnerships for the Sustainable Development Goals: A Case Study of Irish Non-State Actors

**Aparajita Banerjee [1],\*, Enda Murphy [2] and Patrick Paul Walsh [3]**

1   College of Business, University College Dublin, D04 V1W8 Dublin, Ireland
2   School of Architecture, Planning and Environmental Policy, University College Dublin,
    D04 V1W8 Dublin, Ireland; enda.murphy@ucd.ie
3   School of Politics and International Relations, University College Dublin, D04 V1W8 Dublin, Ireland;
    ppwalsh@ucd.ie
\*   Correspondence: aparajita.banerjee@ucd.ie

**Abstract:** The United Nations 2030 Agenda emphasizes the importance of multistakeholder partnerships for achieving the 17 UN Sustainable Development Goals (SDGs). Indeed, Goal 17 includes a target for national governments to promote multistakeholder partnerships between state and non-state actors. In this paper, we explore how members of civil society organizations and the private sector perceive both the possibilities and challenges of multistakeholder partnerships evolving in Ireland for achieving the SDGs. The research uses data gathered during 2018 and includes documentary research, participant observations of stakeholder forums in Ireland and the United Nations, and semi-structured interviews to address related questions. The results demonstrate that numerous challenges exist for forming multistakeholder partnerships for the SDGs, including a fragmented understanding of the Goals. They also note previous examples of successful multistakeholder partnership models, the need for more leadership from government, and an overly goal-based focus on SDG implementation by organizations as major impediments to following a multistakeholder partnership approach in the country. These findings suggest that although Goal 17 identifies multistakeholder partnerships as essential for the SDGs, they are challenging to form and require concerted actions from all state and non-state actors for SDG implementation.

**Keywords:** Sustainable Development Goals; Goal 17; multistakeholder partnerships; challenges of multistakeholder partnerships; stakeholders' perceptions

## 1. Introduction

The interconnected nature of the 17 SDGs outlined in the UN Transforming Our World: The 2030 Agenda for Sustainable Development calls for a 'whole-of-society' and a 'whole-of-government' approach to implement plans, projects, and policies within member states [1]. In other words, Agenda 2030 stresses the idea that state and non-state actors should work together to achieve the 169 targets of the SDGs [2,3]. Goal 17 of the SDGs has indicators that point towards partnerships between these state and non-state actors, indicating that "a successful sustainable development agenda requires partnerships between governments, the private sector, and civil society" [1]. One of Goal 17's quantifiable indicators is "the amount of United States dollars committed to public—private and civil society partnerships". Another indicator specifies that member states report "progress in multistakeholder development effectiveness monitoring frameworks that support the achievement of the SDGs" [1].

However, achieving these targets and indicators warrants multiple intra-sectoral and inter-sectoral entities to work collaboratively with other actors, co-sharing resources, expertise, and responsibilities to address complex and multifaceted social, economic, and environmental problems that are of mutual interest. There are plenty of critical debates on pursuing such an approach, including questions around how partnerships evolve and operate. However, these debates continue without clear pathways of what works or otherwise. Despite this, Agenda 2030 places critical importance on partnership for achieving the SDGs with the UN designating it as one of the "5Ps of sustainable development" (along with people, planet, prosperity, and peace).

Given the importance placed on the need for different groups of stakeholders to work together in implementing the SDGs, the research question explored in this paper is how non-state actors, including civil society groups, environmental organizations, youth groups, trade unions, and business associations in Ireland perceive the feasibility, opportunities, and challenges of forming multistakeholder partnerships? Little is known about how non-state actors envisage working with each other and with partners from other societal sectors for achieving the SDGs; similarly, there is a lack of understanding about the core obstacles and challenges faced by these groups. The objective of this paper is to contribute to filling the knowledge gaps in these areas.

## 2. Background and Literature Review

This section begins with defining multistakeholder partnerships and is followed by a literature review of what scholars have identified as the different benefits and challenges of partnerships for achieving the SDGs. A host of search terms such as 'partnerships', 'multistakeholder partnerships', 'partnerships for the SDGs' was used to explore scholarly articles in databases such as Scopus, Web of Science, JSTOR, and Google Scholar. However, the different benefits and challenges of multistakeholder partnerships discussed in this section is not an exhaustive list and focuses on literature that is germane to our research.

### 2.1. Multistakeholder Partnerships: Definitions and Benefits

United Nations General Assembly Res. 60/214 defines partnerships as 'voluntary and collaborative relationships between various parties, both state and non-state, in which all participants agree to work together to achieve a common purpose or undertake a specific task and to share risks and responsibilities, resources and benefits' [4]. Similarly, academic scholars define partnerships as 'a voluntary cooperative arrangement between organizations from the public, private and/or civil society sectors . . . that have common, non-hierarchical decision-making procedures and share risks and responsibilities . . . to address a public policy issue' [5] (pp. 6–7). Common to both foregoing definitions is a collaborative relationship between the different societal actors intending to address a mutually beneficial public good challenge. As multiple state and non-state actors are involved, these partnerships are defined as multistakeholder partnerships. Given the representation of different sectors, these partnerships can lead to what Severino refers to as 'hypercollective action' [6] (p. 11) which is more inclusive in terms of membership and better suited to solving complex problems [7–9] targeted by the SDGs [10].

Multiple authors have pointed towards a plethora of benefits of partnerships between different societal sectors. Partnerships fill a void created by the government's inability to reach specific societal segments due to either unwillingness or a lack of resources [11]. When different sectors collaborate, scholars have found evidence that partnerships can lead to improved efficiency, cost reduction, and innovation [12–14], and that accessing knowledge, networks, resources, and opportunities can become easier [9,15–17]. Furthermore, co-learning is possible as actors learn from each other [15,18], and the benefits and risks are shared by different partners [19]. New relationships based on trust, reputation, and legitimacy can emerge [15,18], reducing conflicts due to the shared benefits from achieving goals [20].

Given the various potential benefits of forming partnerships, it is unsurprising that the UN espouses them as critical for implementing the 17 SDGs as achieving many of the targets is beyond the

scope, resources, and ability of the government of each member state. As Kanie et al. [3] rightly note, "the theory of change is that once stakeholders sign up, they set priorities, aggregate resources, create the necessary institutions or adapt existing ones, and galvanize people and institutions to pursue the goals" (p. 3).

*2.2. Multistakeholder Partnerships: Limitations and Challenges*

Despite the benefits mentioned above, some scholars have found that partnerships are not easy to form and maintain in the long run; day-to-day operations are challenging to monitor and can be limited in achieving significant results over time [21]. As multiple types of actors form multistakeholder partnerships, day-to-day operations need to be handled to avoid conflicts [22]. This requires skillful orchestration or what Fowler and Biekart [2] refer to as 'interlocutors' who can guide the partners to achieve the purpose of the multistakeholder partnerships and make them accountable [23]. The spatial scale (e.g., local, regional, national) also affects the success of multistakeholder partnerships as local level actors may lack the collaborative capacity to engage with other local actors [24]. As a result, more resourceful organizations can capture multistakeholder partnership processes, delimiting the scope of participation and entry points for smaller-scale organizations, especially those working at the grassroots level. Hence, multistakeholder partnerships may not necessarily result in more inclusive processes as some groups may dominate more than others [25]. Besides, working in a partnership may demand changes in how individual actors operate to achieve their own organizational goals. Therefore, it requires actors to balance both partnership goals and their own goals as an organization [26,27].

The positive connotations of the word 'partnerships' can make these arrangements automatically palatable or attractive [28]. Different groups of actors are expected to work with each other to pull together resources and skills to solve public policy problems as "a more effective way of delivering policy interventions than state-led or 'top-down' approaches" [29] (p. 149). Others suggest that their positioning as a panacea for societal problems often makes their critical examination or challenging the approach taken within partnerships quite problematic [30–32]. Therefore, the 'enthusiasm for partnerships' [30] (p. 307) and presenting or interpreting partnerships solely from a positive angle often limit objective analysis of the implications of partnership [30,32].

There is a considerable amount of research on what works and does not work in partnerships. One strand of research concentrates on the internal arrangements within a partnership that creates conditions for success, whereas other studies have investigated the external or socio-economic conditions under which partnerships operate that influence their success [9,24,27,33,34]. Other research has found that specific public policy issues can bring different relevant and interested stakeholders together [35]. Moreover, partnership success can depend on the inclusion and participation of multiple stakeholder groups [36]. However, at times in such significant partnerships, individual partners may prioritize partnership goals over organizational goals, thus offending the critical players invested in the partnership [24]. Participating in partnerships also requires financial resources and human capacities within each stakeholder group that focuses primarily on the reporting and monitoring related work [37]. Other research has discussed the power dynamics within partnership arrangements whereby more powerful stakeholders tend to exert too much control over processes [38]. Overall, while multistakeholder partnerships are very well suited to the concept of sustainability [34], the general trend in scholarship is to advise caution in assuming that partnerships are panaceas and that rigorous empirical research is required to explore the actual effectiveness of partnerships rather than adopting a normative understanding of the term [31,32].

*2.3. Multistakeholder Partnerships for the SDGs: Complexities and Review of Existing Literature*

The scale, scope, interconnections, and interdependencies of the SDGs require a 'whole-of-society' and 'whole-of-government' approach as governments alone cannot achieve them. It is clear from the above discussion that although partnerships may fit well in many of the execution plans for achieving the SDGs, they are far from being the solution for all implementation-related problems. Therefore,

stakeholders who can form partnerships to implement the SDGs require an understanding of their complex nature.

Several authors have identified the critical importance of aligning the work done in different sectors to achieve the SDGs [39–49]. For example, Rosati and Faria [44] studied how different companies aligned their corporate social responsibility-related activities with the SDGs and found that companies operating in countries with a high level of climate vulnerability are more SDG-aligned. Another study focusing on the private sector's role in developing socially relevant business models points to the importance of social impact bonds (SIB) to achieve such goals. SIBs are a hybrid arrangement between social and financial institutions that creates a unique platform to support public–private partnerships between different sectoral actors working collaboratively to achieve the SDGs [45]. Several other studies take a similar sectoral focus making cases for the SDGs in education [42,43,48], in addressing the needs of women [46], for rehabilitating vulnerable communities [47], across value chains [49,50], and in the banking sector [51].

Though existing research draws attention to what needs to be done, there is also a critical need to understand how different actors can work collaboratively in doing what needs to be done. There needs to be strong and inspiring leadership for bringing diverse groups of actors to align their organizational goals and objectives with the SDGs. Fowler and Beikart [2] propose that interlocutors, meaning 'secretariats, focal points, platforms, hosts and other labels for a critical player' (p. 81), can play the role of an orchestrator setting rules and overseeing the operations of the initiatives undertaken by the partnerships. However, more research is still required to understand how these processes can evolve and how state and non-state actors can provide the most efficient and effective support. In this paper, we explore some of the critical components discussed above in the context of multistakeholder partnerships as a tool for implementing the SDGs in the Irish context.

## 3. Materials and Methods

As the research aimed to understand how various non-state actors in Ireland envisaged the formation of multistakeholder partnerships working collaboratively to achieve the SDGs, qualitative research methods were used to contextualize, interpret and understand the various background perspectives. Purposive document sampling was used to select documents containing rich information [52] on multistakeholder partnerships and why and how the United Nations identified multistakeholder partnerships as important for implementing the SDGs. Documents such as Agenda 21, the Millennium Development Goals, and the 2030 Agenda were useful. Other UN documents such as Voluntary National Reviews of SDG implementation progress were used to acquire preliminary information about how different countries envisage the multistakeholder partnership processes. Participant observation at both the 2018 and 2019 UN High-Level Political Forum also helped identify the types of actors who participate in reviewing the SDG implementation progress globally and from Ireland. Journal articles, reports, and news clippings were explored and provided an in-depth understanding of what kind of non-state actors and sectors can play a critical role in implementing the SDGs, and a list was prepared for such actors in Ireland.

The document research was followed up with semi-structured interviews with 14 key experts and senior members of different civil society umbrella organizations, environmental groups, trade unions, and organizations that promote business interests in local communities in Ireland (Table 1). Most of the organizational representatives interviewed were regular attendees of the national SDG Stakeholder Forum's organized every quarter since 2018 by the Irish Department of Communication, Climate Action, and the Environment (DCCAE) as a platform for non-state actors to interact with the government and to be aware of government initiatives for the SDGs. Representatives of many of these organizations also regularly participated in the UN High-Level Political Forum on the SDGs. We chose interviewees from people who attended the National Stakeholder Forums to identify critical experts aware of plans and progress on the SDGs in Ireland. However, not all interviewees and not all types of non-state actors interviewed were attendees of the National Stakeholder Forums such

as representatives of business organizations and trade unions. A couple of our interviewees were unaware of the SDGs and their scope and scale, even though their organization worked on multiple issues covered by the Goals. For ethical reasons, we will not divulge the names of the organizations whose members were interviewed.

**Table 1.** Interviewee List.

| Non-State Actor Type | No. of Interviewees | Interviewee Number |
| --- | --- | --- |
| Youth organization | 1 | ENGO 001 |
| Civil Society Organization | 3 | ENGO 002, ENGO 003, ENGO 004 |
| Environmental NGO | 3 | ENGO 006, ENGO 007, ENGO 008 |
| Community organizations | 2 | ENGO 005, ENGO 009 |
| Business organizations | 2 | BIZ 001, BIZ 002 |
| Trade unions | 3 | ENGO 010, ENGO 011, ENGO 012 |

Non-probability purposive 'snowball' sampling methods were adopted to generate respondents from each of the sectors outlined in Table 1. One issue that we considered in the sample generation is that respondents suggest other potential interviewees who share similar characteristics and outlooks. In such a case, it is essential to ensure that the respondents meet established screening criteria to reduce the possibility of bias developing in the sample [53]. This was particularly important when respondents suggested other people 'who might be worth talking to' [54]. As a result, a core qualifying criterion was established, irrespective of interviewees' referrals, to enter our sample, respondents had to be a senior member of their organization and currently be in a leadership role. Bearing this in mind, a quota of 3 respondents was sought from each sector shown in Table 1, equating to 18 respondents in total. However, while 23 interviewees were targeted, only 14 agreed to participate in the study. Despite this, Mason's [55] survey of 2533 studies that employed qualitative approaches found that small sample sizes are standard in studies using qualitative methods, and therefore, we consider the current sample adequate for meeting the study objectives.

A single protocol pre-approved by the university research ethics board was used for conducting all semi-structured interviews. However, follow-up questions were also asked of interviewees that were specific to their sector or work area. For example, many of the follow-up questions asked of business organizations were different from that of civil society organizations. After initial warm-up questions, the protocol consisted of questions to understand how interviewees perceived the concept of sustainability, their knowledge of the SDGs, how their work aligns with SDGs if at all, whether or not other members of their organization were aware of the SDGs, their views on multistakeholder partnerships, the opportunities and challenges they envisaged in forming multistakeholder partnerships, what role they felt the government should play in building partnerships, and how they perceived Ireland's progress in achieving the SDGs.

We acknowledge that undertaking qualitative interviews can present methodological limitations. In this regard, we were cognizant, in particular, of the gaps between what was said in the interview setting and what occurred in reality [54]. Dunn [56] warned of the dangers of the 'pufferfish' phenomenon, where respondents (particularly those in positions of authority) attempt to portray themselves or others in a particular light, and this was regarded as a real issue for our research. Several steps were taken to help ensure that the respondents offered transparent and frank responses. First, the interviews were anonymous to encourage the respondents to be as open and transparent about their experiences as possible. Second, considerable attention was taken to ensure that the respondents felt comfortable with the interviewer. Professionally formulated emails were issued to prospective respondents, which set out how the interview information would be gathered and used. Respondents were also informed that the interviews would be recorded digitally and transcribed and that their organization's anonymity would be protected. They were also assured that the data generated would be used solely for independent academic research purposes.

All interviews were transcribed verbatim, and a systematic in-depth review of the interview transcripts was then carried out on a line-by-line basis to develop codes that were used to sort the data. Given that the questions were mainly organized around themes, the coding was straightforward, and common themes were easily identifiable. To understand the most common themes emerging from the data, and their prevalence among the sample, the number of respondents who raised particular themes/codes were documented and quantified. This meant we could get a clearer picture of the pervasiveness of dominant issues relating to SDG awareness for multistakeholder partnerships, the history of partnerships, intersectoral relationships between non-state actors, and the government's potential role. NVivo 12 software was used for coding the data.

## 4. Results

Though multiple other themes emerged from the research on the progress of the SDGs in Ireland and the different opportunities and challenges, the results presented in the section were focused on the key objectives of the paper, consisting of (1) understanding the role of SDG awareness or otherwise for facilitating multistakeholder partnership formations; (2) exploring the role of past/existing institutional knowledge and memory in multistakeholder partnerships formation; and (3) understanding the role of government for multistakeholder partnership formation for the SDGs.

### 4.1. SDG Awareness and of Importance of Partnerships

The first theme that emerged from the interviews was that SDG awareness in general in Ireland is low among non-state actors across different spatial scales (local, regional, national). Because we interviewed non-state actors operating at the national level, they provided insights on the awareness of organizations under their umbrella (i.e., local and regional levels). Apart from intra-organizational awareness, interviewees also pointed out that the community-level knowledge of the SDGs was low. Even when interviewees were aware of the Goals, they had little knowledge of the 169 targets and 231 indicators. Therefore, it was not surprising that they also had little awareness of the focus on partnerships as a means of SDG implementation identified in Goal 17. A total of 11 of 14 interviews referred to SDG awareness as a challenge in multistakeholder partnership formation. Even when the awareness of the SDGs and the role of partnerships for their implementation was evident, it was confined to the national-level umbrella organizations. According to the interviewees, regional and local level non-state actors were mostly unaware of the SDGs. Where awareness did exist, an in-depth understanding of the complex interconnections of the goals, their targets, and their indicators was lacking. Apart from institutional-level awareness, individual awareness was also low. Indeed, some interviewees were critical of the SDGs' broad scope and complexities and were unaware of how their organization's work aligned with the Goals. As one interviewee asserted:

> *"I mean they are there in the popular opinion, lots of marketing, lights up social media, fine,* [sic] *. . . on the ground, the awareness of SDGs is very poor in Ireland; communities do not know what they are . . . and they're quite complex to explain to somebody, 17 SDGs. I come into a community, and you have to explain like 17 things. What do they take home? Which one? Which SDG?"* (Interviewee ENGO 001)

Given the lack of awareness and the broad scope and scale of the Goals, some interviewees pointed out that they had found ways and means to navigate them by choosing the goal that aligned well with their organization's aims and interests. For example, business organizations and trade unions identified with Goal 8—Decent Jobs and Economic Growth and Goal 12—Sustainable Production and Consumption. Environmental groups identified with Goals 13–15 because those were most directly related to their scope of action. This was evident in the following responses:

> *"We looked at the 17 goals, and we chose one primary goal and four secondary goals . . . A lot of other ones we still identified with [sic], but you can't be a champion [on those]. If you do not do a filtering exercise, the messages get lost"* (Interviewee BIZ 002)

*"We would work with the Climate Action one. I think it's [sic] SDG 13; we would work with the education one, which would be SDG 4. We would work with life on land, life at sea, sustainable production and consumption. We would also link in a bit with the sustainable cities side"* (Interviewee ENGO 001)

Several interviewees also found cherry-picking the goals problematic, given the strong interconnections and interdependencies among the Goals. For them, the overall Goals were more significant than the sum of their individual parts. As one interviewee described:

*"Things are going to change when it starts to affect people directly; if climate change is affecting people, action will be taken if water [sic], is polluted action will be taken; however, there is no one with the vision to see the whole picture"* (Interviewee ENGO 004)

### 4.2. Historical Lack of Multistakeholder Partnerships

The results from interviews also highlighted the distinct lack of knowledge among respondents of multistakeholder partnerships in operation in Ireland—the types of partnerships that the UN recommends for the effective implementation of the SDGs. Interviewees were unable to provide specific partnership examples where they had worked together with multiple entities from civil society organizations, environmental groups, academia, business entities, and other stakeholders to solve a critical public policy problem. Most interviewees identified dyad types of partnerships comprising only two sectors [57] such as government and the private sector or government and civil society working together. As a result, the interviewees demonstrated a distinct lack experience of working in multistakeholder partnerships and what it entails.

However, interviewees who were deeply engaged with the SDGs found value in multistakeholder partnerships as areas where the work of organizations often overlapped or complemented each other. As one interviewee pointed out: *"The SDGs created possibilities for linkages between organizations in different sectors where maybe we wouldn't have thought about those linkages before . . . it has created real opportunities for us to kind of maybe come together"* (Interviewee ENGO 002). For them, the SDGs provided a reason to come together and to collaborate. Some interviewees also pointed out that the complexities in the SDG targets and indicators made it necessary to collaborate, a rallying point for creating multistakeholder partnerships.

Along similar lines, most interviewees noted the potential benefits of more wide-ranging partnerships, including bringing different skills and resources to deliver common goals. Partnerships would be beneficial for small and medium-sized organizations whose resources are limited but work in similar critical social, economic, or environmental areas. As one interviewee pointed out:

*"You should be looking at something where you have more of a symbiotic kind of relationship; that you're bringing something that they don't have and they're bringing something that you don't have and, actually, together you're actually doing something bigger"* (Interviewee ENGO 001)

However, although interviewees generally identified the benefits of multistakeholder partnerships for achieving the multiple complex SDG targets, most interviewees believed it could be challenging and a *"long and messy process"* (ENGO 008). They mostly identified trust, communication, and the partnerships' day-to-day operations as the main challenges for successful multistakeholder partnerships to emerge.

In terms of trust, interviewees believed multistakeholder partnerships should have a clear purpose of what is to be achieved and a plan of how it can be achieved. They also believed that transparency was required regarding who the partners were, why the partnership was being formed, the benefits for individual partners, and their roles and obligations. Trust between all the stakeholders in a partnership was considered crucial given that different stakeholder groups operated under different sets of rules, and as one interviewee pointed out, *"[our] styles of understanding of the world is different"* (ENGO 003). For example, an interviewee from a civil society organization was apprehensive about

partnering with businesses that operated on a mindset of pursuing *'indefinite growth'* (ENGO 008) when they felt such models might not be feasible for achieving the SDGs. On the other hand, another interviewee representing business organizations was apprehensive about collaborating with civil society organizations given that it was *"very difficult to have a proper conversation with them"* (BIZ 002). Interviewees from civil society organizations were also concerned that businesses with more resources could capture multistakeholder partnerships and operate them to suit their purposes.

Some interviewees also pointed out the need for excellent communication for multistakeholder partnerships to flourish. They considered communication as crucial both between partners and within partnerships and that a clear memorandum of understanding should be in place to define the partner roles and expectations in the partnership.

Members of civil society and environmental organizations operated on limited budgets, mostly donor-funded for specific projects, and found it challenging to have resources to concentrate on forming, maintaining, and delivering on partnerships. Additionally, most interviewees believed that the organization's culture sometimes made it impossible to participate in partnerships that would require tweaking or realigning their organizational goals and values. These factors inhibited organizations from experimenting with partnerships.

### 4.3. The Role of Government

Given that interviewees lacked the experience working in multistakeholder partnership-type arrangements, they generally believed that the government should initiate and play a critical role in encouraging and promoting partnerships in the country. Most interviewees believed that as the government is answerable to the UN to fulfil the Irish SDG commitments, it should steer the process of achieving the Goals and should involve creating opportunities for non-state actors when and where required. Government institutions and policies could set conditions that would facilitate *"the trade-offs and the sacrifices people have to make to be in a partnership"* [ENGO 002].

Interviewees representing the private sector felt that the SDGs provided new entry points in public governance, where further public–private partnerships could be organized to deliver socially impactful projects. As a result, they perceived that it was the government's role to encourage increased public–private partnerships in SDG-related projects. They felt that such projects could also create new types of jobs for a more sustainable economy. However, they were also of the view that the right opportunities and processes were not yet in place in Ireland for the private sector to create innovative solutions that would drive sustainable processes like a circular economy, improved waste management, and other green initiatives. They felt that to stimulate such processes, government business development agencies could provide funding for projects aligned with SDG targets and simultaneously create opportunities for the business sector.

Additionally, business sector representatives pointed out that the business sector's role in meeting SDG targets is limited to their corporate social responsibility (CSR) activities. However, according to them, SDGs' scope was too large to be dealt with via corporate social projects alone. The SDGs covered wide-ranging goals like climate action, poverty alleviation, and biodiversity loss, which required systemic changes within industries and that government intervention would transform. To signal industries to move towards sustainable business practices, the interviewees from the business sector suggested that the government initiate green procurement programs as supplying goods and services to the government constitutes a significant part of private sector business activities.

Interviewees representing civil society and environmental organizations were also of the view that the government needed to play an active leadership role in SDG implementation. Most interviewees suggested that implementing the SDGs should become the head of the state's priority in Ireland. Most civil society and environmental groups did not perceive any shrinking of the space for civil society in Ireland like in some other parts of the world, and that scope for collaborative work with the government and others on implementing the SDGs could be developed. However, most interviewees were concerned that the government is dominated by the private sector that has packaged social and

environmental commitments covered under the SDGs into small and neat CSR packages that do not do justice to the complex issues at the root of sustainability problems.

Interviewees with extensive experience working in grassroots and community-level organizations pointed out that there were significant challenges for the government to engage with organizations that worked with those furthest behind. Many of these organizations may work in remote locations and, as a result, find it impossible to enter into regular dialogue with the government, including participating in the quarterly held National SDG Stakeholder Forums. Because of this, they felt that SDG implementation planning should also be focused on the regional and local levels so that more social groups can participate and co-share responsibilities for achieving the SDGs.

Although all interviewees believed that the government has a major role in implementing the SDGs, most of them felt that there is currently a lack of political will in the national government to pursue the SDGs. As one interviewee said, a *'re-ignition'* (ENGO 004) is required. The SDGs' scope and scale require the leadership of the national government given that the systemic changes needed to implement the SDGs are beyond the scope of local government. One interviewee from the private sector said, " … *I don't think that the Government's* [sic] *doing enough to put the right tools in place to facilitate businesses"* (Interviewee BIZ 001). Another civil society interviewee pointed out:

> *"there is a lot of tokenism going on in the name of SDGs; there must be more meaningful engagement where figures need to be robustly proved through validation and verification, and the government should stand up on their heels and absolutely honour the SDGs"* (Interviewee ENGO 005)

However, although interviewees were critical of what they viewed as a lackadaisical attitude of the government towards the SDGs, most interviewees pointed out that although issues like climate action, watershed management, biodiversity loss, and ocean health are essential to the sustainability of Ireland, political leadership is now compelled to concentrate on broader geopolitical issues such as Brexit.

## 5. Discussion

Our research uncovered some critical insights from the stakeholders on the SDGs, multistakeholder partnerships, and the government's role that may be unique to Ireland. However, many of these findings resonate with existing research on multistakeholder partnerships and problems identified by other scholars in implementing the SDGs. We also found empirical evidence of how non-state actors envision real-life challenges and obstacles in forming multistakeholder partnerships for the SDGs. These are now discussed.

### 5.1. Fragmented Understanding of the SDGs and Cherry-Picking of the Goals

The general view from the results suggests that there are too many goals and too little awareness of the goals. The results highlight that the broad scope of the Goals, the complexity of their interconnectedness, and the various scales at which actions must be taken is yet to be fully grasped by non-state actors in general. Such a lack of awareness of the detail of the Goals can act as an impediment for non-state actors who may be the future agents of change. Not surprisingly, they cherry-pick from the Goals aligning with those best suited to their organizational goals and tend to pursue the goals separately, within silos, even though the SDGs are inherently interlinked. We also argue that this lack of awareness can also act as inertia for non-state actors to better align their work with the SDGs. This issue is not specific to the Irish context and has been outlined in previous research [58].

Cherry-picking Goals that fit with organizational narratives tend to promote the continuation of silo-thinking. Other researchers have also identified the cherry-picking of the Goals at larger scales, for example, within nation-states where some nations prioritize poverty alleviation and economic growth over other Goals [59]. We argue that such fragmentation of the Goals may create challenges for organizations to take a holistic view of the SDGs and their core principles of universality and

indivisibility. Moreover, some Goals are at risk of being ignored, especially the environmental goals when economic and social goals are prioritized.

The lack of awareness of the Goals within non-state actors operating at different levels with a low level of awareness at local and regional levels is also critical. With change agents working in communities unaware of the Goals, the whole-of-society approach required for the Goals can be limited. This finding is not unique to our research and has been outlined by other scholars who point out that SDGs' awareness continues to be low in different sectors [60–62]. A lack of awareness among citizens and citizen groups is problematic as awareness acts as a prerequisite for policy acceptance creates pressure on policymakers to implement specific policies over others [63]. However, awareness campaigns and training programs can solve this problem to a large extent. The future generation of policymakers, corporate workers, social actors, innovators, and citizens educated on the Goals and their ideals can solve this problem if the right measures are put in place.

### 5.2. Challenges of Multistakeholder Partnerships

Our results suggest that well-functioning multistakeholder partnerships that effectively deal with different challenges covered under the Goals and their targets have a long way to go. Again, this is not specifically a problem in Ireland. The existing literature suggests that there is still much to be understood about creating an ideal model or template of multistakeholder partnerships [7–12].

Although empirical research on how stakeholders perceive participating in multistakeholder partnerships for the SDGs is uncommon, our findings are similar to other studies that have found that multistakeholder partnerships are challenging to form and maintain [21], require skilful facilitation to show results [7,23] and that smaller organizations that work at the local level or with minority interest groups may lack the resources needed to participate in multi-sectoral partnerships [24]. We also found that, like other studies, different non-state actors are apprehensive of power capture within multistakeholder partnerships [25], especially when dealing with the private sector. This portends trust deficits between stakeholders in multistakeholder partnerships that require accurate multi-sectoral representation. At times, organizational legitimacy is more critical to stakeholder groups than that of the partnership goals, affecting the balance required between partnership goals and organizational goals for multistakeholder partnerships [13,21,27].

Nonetheless, interviewees were unable to point towards successful multistakeholder partnerships in Ireland working on public issues; what currently exists in Ireland in terms of partnerships are dyad types of partnerships like public–private partnerships, long championed in Irish national development plans [64]. Although they have achieved significant progress, these dyad-type public–private partnerships have also been highly problematic [64–68]. However, many things can be learned from them to inform how multistakeholder partnerships can be designed and what to avoid in the future. Many of the concerns with public–private partnerships are reflected interviewees' apprehension on multistakeholder partnerships. They raised concerns about the effectiveness of partnerships, how they can be made more accountable, and how to maintain a high level of trust and accountability within the partnership. Similar concerns have been expressed by other researchers studying multistakeholder partnerships [7–12].

A study of a decade of public–private partnership projects in Ireland has shown that such arrangements may not be an effective way to use taxpayers' money [69]. Public–private partnerships also have high requirements for monitoring, accountability, supervision, performance management, and relationship management during the tenure of a contract [69]. Our interviewees also raised such apprehensions when they expressed opinions related to trust, communication, division of roles, accountability, and compliance. Maintaining and delivering on standards set on these criteria also requires time and resources. Moreover, there is also a temporal element. Many of the complex challenges required to be addressed to fulfil the SDGs cannot be unpacked in small packages and require years of projects and continuity. This also means that partnerships need to be pursued over the long term—years, if not decades—to deliver success. A high level of trust and reciprocity is required

to sustain such arrangements, which is difficult to build or predict at the early stage of partnership formation when agreements are put in place [70].

What emerges from these findings is that perhaps multistakeholder partnerships should not be pursued for the sake of it; they need to grow organically based on relationships of trust and accountability. However, for that to happen, perhaps the foremost requirement is how non-state actors are invested in the visions of the SDGs and what are opportunities for cross-sectoral engagements.

### 5.3. Role of Government

What became apparent from the interviews is that the non-state actors felt that the national government must play a vital role in encouraging and promoting partnerships, similar to what is reflected in the targets on partnerships under Goal 17. Other scholars have also identified this and suggested that governments should play the orchestrator's critical role (the 'interlocuter'), of the Goals [7]. Governments need strategies to design policies and plans in an integrated manner, aligning with the SDGs, and overhauling the status quo [71,72]. This would also require coordination between government departments and levels, both horizontally and vertically managed over time so that any spillovers and trade-offs are handled effectively [71].

However, according to an interviewee, what was missing was a clear indication of how different non-state actors could identify themselves as change agents, enter collaborative arrangements within and across sectors, arrange for resources, and deliver priorities. Though this is not entirely missing, it is mostly within a particular sector or corporate social responsibility type project where the private sector works with civil society organizations on small, limited-time projects. Though the government needs to play a more significant role to signal that the SDGs are priorities that need to be honoured, non-state actors also need to become agents of change [73]. Businesses can co-fund projects while civil society organizations can support governance [71]. However, it remains to be seen how both state and non-state actors galvanize into more significant action or when and how the "re-ignition" (ENGO 004) happens.

## 6. Conclusions

The 17 UN Sustainable Development Goals focus on improving human well-being and prosperity. However, for that to happen, a whole-of-society approach is required where non-state actors participate and play an effective role in delivering transformative change. This expectation is enshrined in the Goals, with Goal 17 having targets for national governments to encourage multistakeholder partnerships where different non-state actors can participate in collaborative work to achieve the SDGs. In this paper, we took the case of Ireland and explored how non-state actors perceived the idea of multistakeholder partnerships and their associated challenges and obstacles.

This research has three broad conclusions. First, there is still a significant lack of awareness of the SDGs in Ireland and among non-state actors. Although non-state actors operating at the national-level or based in the country's capital were more aware of the SDGs, their counterparts at local and regional levels had far less awareness of the Goals. Furthermore, community-level awareness was also reported to be low. This negatively affects the whole-of-society approach that is required to achieve the SDGs. A low level of awareness about the Goals' indivisibility and universality also affected how non-state actors envisioned the Goals, often identifying and championing those Goals that fit well with their organizational objectives. Second, there was also a lack of examples of how multistakeholder partnerships have worked in governance. Third, the Irish government must play a more significant role in implementing the SDGs and galvanizing different non-state actors to co-share the responsibility of achieving the SDGs. Indeed, there is a widespread expectation that the government should facilitate multistakeholder partnerships and act as an interlocutor or an orchestrator [7] in their implementation.

Based on our study in Ireland and how some civil society, environmental groups, trade unions, and business organizations are organized in the country, we recommend some solutions. Given that most regional and local social and environmental organizations, as well as small and medium

businesses are affiliated to some form of national organization, these national-level organizations can find solutions on how to participate in multistakeholder partnerships and collaborate from their networks in different partnerships. They can also address resource capacity problems by employing central resources to oversee the day-to-day engagement of multistakeholder partnerships and protect the interests of the organizations participating in their network. As these national organizations are membership-based, the operational cost of these new resources could be funded through very marginal increases in membership fees. Therefore, local organizations having memberships in the national umbrella organizations can also participate in multistakeholder partnerships at different scales that align with their goals and purposes. Additionally, the national government that currently engages with stakeholders via the publicly held National Stakeholder Forums can use these forums to deliberate on issues where partnerships can be formed and involve non-state actors acting in the role of an orchestrator. The Irish government has begun to address some of the problems associated with a lack of communication between non-state actors by forming working groups within the National Stakeholder Forums. However, time will tell whether or not these groups are successful in helping form effective multistakeholder partnerships. Future studies could further explore the feasibility and challenges of such an approach.

**Author Contributions:** Conceptualization, A.B., P.P.W. and E.M.; methodology, A.B.; validation, A.B., E.M. and P.P.W.; formal analysis, A.B.; investigation, A.B.; resources, E.M. and P.P.W.; data curation, A.B.; writing—original draft preparation, A.B.; writing—review and editing, E.M.; supervision, P.P.W.; project administration, E.M.; funding acquisition, P.P.W. and E.M. All authors have read and agreed to the published version of the manuscript.

**Funding:** This research was funded by Environmental Protection Agency, Ireland under the EPA Research Programme 2014–2020.

**Acknowledgments:** The authors would like to thank the EPA Research Programme, a Government of Ireland initiative funded by the Department of the Environment, Climate and Communications. It is administered by the Environmental Protection Agency, which has the statutory function of coordinating and promoting environmental research. Thank you to the editor and the two anonymous reviewers for their helpful and constructive feedback that improved this manuscript. Thank you to the interviewees for their time and insight.

**Conflicts of Interest:** The authors declare no conflict of interest.

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
