# Peer review of "Perceptions of Multistakeholder Partnerships for the Sustainable Development Goals: A Case Study of Irish Non-State Actors"

_sustainability, doi:10.3390/su12218872_

Round 1
Reviewer 1 Report
The paper analyses the potential feasibility and perceptions of developing Goal 17 in Ireland. The content is attractive and the research is relevant. However, I recommend that some essential changes are made before its possible publication.
References. The guidelines are not followed and the numbers on the reference list do not correspond to the order of citation. It needs to be completely adapted. I recommend using a reference manager (Mendeley, Endnote...).
Abstract. It is too long and ambitious. I would shorten the first part and simply start from line 23 onward: "The purpose of this paper..." adding more information about the findings.
Introduction. The first part of the introduction is correct. However. I would shorten the introduction, where I would put a brief summary of the state of the art, the gaps that allow for further exploration of the 17 UN SDGs, and clear explanations of the research questions. Later, I would create a new section called Literature background or something similar where I would begin to develop what is currently in point 1.1 of the introduction. Currently, the introduction is 4 pages long! This is not possible.
Literature background (Assuming that the authors follow my previous advice and divide the introduction, creating a new epigraph from 1.1.). The background is very scarce. Not many studies are shown that have worked on this topic. Goal 17 has been worked before in Sustainability (for example, in Méndez-Suárez et al. 2020). Much more work needs to be done in this section. The authors should show previous studies to explain what are the gaps to be filled. On the other hand, I find it very strange that there are no propositions (since this is a qualitative study) or at least no research questions. After analyzing the background, it should be clear what those propositions or research questions are.
Materials & Methods. Although anonymity is desired, a table with the basic characteristics of the 14 interviews should be added (see example in Fornés et al. 2019). The reader then could see if the general profiles are adequate. On the other hand, it is not explained the script and main topics of the interviews, nor how the coding process of the answers was done, nor how the analysis was done (with software or not), the creation of categories, etc. It is necessary to explain these parts.
Results and Discussion. There is no point in joining the two sections. I have never seen it. You should create a results section that shows the main results (I recommend grouping it by research questions or categories). The results are well placed under headings that I understand to be the categories. Only in 3.1 (line 278 onward) there is an interpretation that should be placed outside the results section and put it at the end in a conclusions section. Right now you mix results and discussion.
The authors should take the discussions out of the results section and create a new section where they put the discussion. They can use the comments added in 3.1 and keep part of what is currently in the conclusion section. In other words, you should create a new section that discusses with the previous literature and shows, theoretically, what the contribution of their results.
Conclusion. I would keep a conclusion section in which you clearly state which are the propositions and/or the main findings. All this, in a more direct way than there is now.
I believe that if all this is done, the article could be published and would provide insights about the management and implementation of the SDGs. Good luck.
References.
Méndez-Suárez, M., Monfort, A., & Gallardo, F. (2020). Sustainable Banking: New Forms of Investing under the Umbrella of the 2030 Agenda. Sustainability, 12(5), 2096. https://doi.org/10.3390/su12052096
Fornes, G., Monfort, A., Ilie, C., Koo, C. K. (Tony), & Cardoza, G. (2019). Ethics, Responsibility, and Sustainability in MBAs. Understanding the Motivations for the Incorporation of ERS in Less Traditional Markets. Sustainability, 11(24), 7060. https://doi.org/10.3390/su11247060
Author Response
Dear Reviewer, Thank you very much for your detailed comments and we have tried to address every single one of them as best as we could. Please find them addressed in a tabular form in the attached document. With many thanks, the authors.

Reviewer 2 Report
- The whole structure of the paper needs attention and the usual rule (introduction-rationale-need for the work/research gap and relative research questions-objective, background-literature review, approach-methods-hypothesis development, analysis performed-outcomes, discussion and then a separate section of fruitful conclusions/concluding remarks) could be followed more closely to facilitate the flow of the paper. The abstract should be rewritten in order to communicate clearer the aims/objectives/implications/main outcomes of the study. Please develop further your discussion by drawing on relevant studies and in relation with prior MDPI/Sustainability literature - develop further and expand your final section of concluding remarks; the research and/or policy recommendations in the final conclusion section could be further developed. Cite (primarily) in these final-most critical sections of your manuscript relevant papers published in the Journal you submitted your work to (in order to provide some sort of continuity of the specific research string).
- More references to recent & relevant literature/conceptual and/or empirical studies could increase the quality of the research paper and provide a much clearer message to the reader - these may help you building your discussion which needs to be extended. Add the following to your reference list: Avrampou, A., Skouloudis, A., Iliopoulos, G., & Khan, N. (2019). Advancing the sustainable development goals: Evidence from leading European banks. Sustainable Development, 27(4), 743-757.
- Concluding remarks: authors must elaborate more on what is their contribution to the literature as well as on opportunities for future research. Questions that need to be answered: Why your study is important? and how it extends existing knowledge on the specific issue/topic? Conclusions need to be written in a clear and coherent manner and draw the main lessons from the paper. I suggest you to concentrate on the description of the implications of the work, the main findings and its potential replicability - empirical investigation to other national terrains and settings. Furthermore, limitations of the study need to be outlined to a greater extent, and so are any potential connections between your study and specific aspects of the Journal's scope.
- Carefully check the references, so as to make sure they are all complete and follow the MDPI/Sustainability's Guidelines to Authors in detail.
- Finally, do check thoroughly, in order to avoid grammar, syntax or structure/presentation flaws. Make sure you retain a formal/academic-specific style of presenting your work throughout the text - if necessary, please seek for professional English proofreading services or ask a native English-speaking colleague of yours in order to refine-improve the English of your paper.
Author Response

(The authors gave the same response as above.)

Round 2
Reviewer 1 Report
Congratulations. The changes have been incorporated and I think the result is suitable for this journal.
Author Response
Thank you for your comments
Reviewer 2 Report
The author(s) have adequately responded to comments and recommendation made in the first round of reviews.
Author Response
Thank you for your comments, English language check conducted.